# Targeting Autophagy to Overcome Human Diseases

**DOI:** 10.3390/ijms20030725

**Published:** 2019-02-08

**Authors:** Maria Condello, Evelin Pellegrini, Michele Caraglia, Stefania Meschini

**Affiliations:** 1National Center for Drug Research and Evaluation, National Institute of Health, Viale Regina Elena, 00161 Rome, Italy; maria.condello@iss.it (M.C.); evelin.pellegrini@guest.iss.it (E.P.); 2Department of Precision Medicine, University of Campania “Luigi Vanvitelli”, 80138 Naples, Italy

**Keywords:** autophagy, cell survival, cell death, target therapy, inflammation, metabolic diseases, neurodegenerative diseases, autoimmune diseases, aging, cancer

## Abstract

Autophagy is an evolutionarily conserved cellular process, through which damaged organelles and superfluous proteins are degraded, for maintaining the correct cellular balance during stress insult. It involves formation of double-membrane vesicles, named autophagosomes, that capture cytosolic cargo and deliver it to lysosomes, where the breakdown products are recycled back to cytoplasm. On the basis of degraded cell components, some selective types of autophagy can be identified (mitophagy, ribophagy, reticulophagy, lysophagy, pexophagy, lipophagy, and glycophagy). Dysregulation of autophagy can induce various disease manifestations, such as inflammation, aging, metabolic diseases, neurodegenerative disorders and cancer. The understanding of the molecular mechanism that regulates the different phases of the autophagic process and the role in the development of diseases are only in an early stage. There are still questions that must be answered concerning the functions of the autophagy-related proteins. In this review, we describe the principal cellular and molecular autophagic functions, selective types of autophagy and the main in vitro methods to detect the role of autophagy in the cellular physiology. We also summarize the importance of the autophagic behavior in some diseases to provide a novel insight for target therapies.

## 1. Introduction

Deregulation of autophagy is involved in the pathogenesis of a wide range of human diseases. Herein, an overview of the mechanisms regulating cellular autophagy has been developed to elucidate the biological processes that allow cells to self-eat, in a continuous recycle of renewal, eliminating the harmful or useless parts. Autophagy is a conservative process and plays an essential role in maintaining and regulating cellular balance and physiology. Autophagy regulation is arranged in a systemic way for the degradation of altered organelles and abnormal proteins. Autophagic process alteration producing a malfunction may be the cause of diseases, such as tumors, metabolic dysfunctions, neurodegenerative and inflammatory diseases [1]. Most recent scientific research shows that the breast and ovarian cancer are associated with Beclin1 gene mutation [2]; Crohn’s disease is related to the mutation of the *ATG16L1* gene [3]; the polymorphism in *ATG5* gene is associated with asthma [4]; the polymorphism in *ATG5* gene and HRES-1 locus with systemic lupus erythematosus [5,6]; p62 protein mutation is associated with amyotrophic lateral sclerosis [7]; mitophagy deficiency due to the PINK1 mutation is associated with Parkinson’s disease [8]; diabetes mellitus is also related to dysregulation of autophagy which induces an alteration in the normal function of pancreatic β cells and the complication of insulin resistance in this disease [9]. The early inactivation of autophagy may induce transformation of normal cells into tumor cells. When the microenvironment is metabolically unfavorable to tumor cells growth or when tumor cells are being treated with toxic drugs, the activation of autophagy gives to the cells an advantage for survival [10]. Understanding the role of autophagy in the pathogenesis is very important for the therapeutic strategy evaluation and for improving every individual treatment to overcoming diseases [11].

## 2. Cross-Talk between Autophagic, Apoptotic and Necrotic Pathways

Autophagy is a catabolic process that allows cellular homeostasis maintenance through the elimination of unnecessary proteins and damaged organelles. This process is activated under physiological conditions (lack of nutrients, deprivation of growth factors) or in response to a variety of different stress stimuli (deficiency of oxygen, the induction of oxidative stress or toxic agents exposure). This phenomenon is important for the cell by ensuring the maintenance of the energy balance, it avoids that also small energetic variations can be detrimental to the cell itself. The basal autophagy cellular levels, i.e., those present even in the absence of stimuli, allow the normal turnover of proteins and organelles, through the elimination of superfluous organelles and non-useful aggregated proteins. This process also allows cells to generate energy when nutrients are scarce and provides bioenergetic support during development (survival response). If autophagy is inhibited, accumulation of intracellular damaged organelles is observed; protein aggregation and deficiency of the correct energy supply bring to cell death. With an uncontrolled autophagic cellular process, the vital cellular organelles and the useful biological macromolecules are degraded, the antiapoptotic factors are digested with consequent interference with the physiological survival mechanisms, which can cause the death of the cell itself, i.e., autophagic cell death. Some studies have reported that, during *Drosophila* development, autophagy was involved as a physiologically relevant model of cell death, whereas apoptosis and necrosis had not activated [12]. However, the term “autophagic cell death” should be used with caution and moderation. Since autophagy is a typically pro-survival process, to prove that it has a causative role in cell death, it is necessary to have adequate evidence [13]. We must prove that other mechanisms of cell death are not responsible and verify that inhibition of autophagy, by genetic or chemical means, prevents cell death. Moreover, morphological criteria alone are not sufficient to assess cell death; it is important to use more than one assay to measure cell death and, in particular, perform cellular viability assay [14].

The mechanisms regulating several kinds of cell death, such as apoptosis, autophagy and necrosis, are often separate, but occasionally they overlap and can be activated simultaneously or subsequently in response to an excessive stimulus received from the cell. Many researchers have underlined a connection between autophagy and apoptosis [15]. In the same cell, apoptosis or autophagy pathways can occur simultaneously or in sequence in response to the same stimulus, either when the autophagy is protective or cell death inducing. After a stress signal, the cell activates survival autophagy that inhibits apoptotic cell death (Figure 1A). In other cases, the autophagic process cannot defend the cell from the damage induced by stress and dies by apoptosis (Figure 1B). Alternatively, autophagy and apoptosis work together and cell dies for both mechanisms (Figure 1C). The subcellular compartments where autophagy and apoptosis could be interconnected are the endoplasmic reticulum [16], mitochondria [17] and lysosomes [18].

The cross-talk between autophagy and apoptosis is due to the damage extent and the cell susceptibility. There are multiple signaling pathways independently controlling different types of cell death. When the cells are stimulated by stress, autophagy is induced prior to apoptosis, whereas apoptosis and necrosis are induced if autophagy is inhibited or ineffective. The main characteristics of apoptosis, necrosis and autophagy are summarized in Table 1.

## 3. Molecular Regulation of Autophagy 

Autophagy is a dynamic and multi-phase process, as described in Figure 2; when a cell is undergoing autophagy, a small double layer membrane (phagophore) is formed (Figure 2A), which begins to wrap the organelles and/or the macromolecules to be eliminated (Figure 2B). This membrane closes, forming the autophagosome, i.e., a double membrane vacuole (Figure 2C). The autophagosome merges with the lysosome (Figure 2D) to form the autophagolysosome (Figure 2E), in which the hydrolytic lysosomal enzymes digest the inner membrane of the vacuole and the autophagosomal content (Figure 2F). The macromolecules released can be used for biosynthesis or to supply cellular energy [19].

Tests of genetic screening on the yeast *Saccharomyces cerevisiae* made it possible to understand the molecular regulation of autophagy, thanks to the identification of about 30 genes, whose protein products are able to control autophagy (called precisely *ATG* genes, AuTophaGy related genes) [20,21]. Most of these genes have homologous sequences in higher eukaryotes, from fruit fly to mammals, suggesting that the molecular mechanism of autophagy is highly conserved in evolution [22]. 

When nutrients are available, the mammalian target of rapamycin (mTOR) and protein kinase A (PKA) negatively regulate autophagy through the phosphorylation and inhibition of the induction complex, called unc-51-like kinase 1 (ULK1) complex [23]. When nutrients are scarce, different chemical mediators are released by stress as AMP-ATP ratio rises, there is elevated production of reactive oxygen species (ROS), nicotinamide adenine dinucleotide (NAD^+^) increases, there is inhibition of mTOR activity, and switch-on of AMP-activated protein kinase (AMPK), initiating the autophagic machinery (Figure 3). The induction phase of phagophore formation is mediated by the ULK1 complex, which includes ULK1, ATG13, focal adhesion kinase family interacting protein of 200 kDa (FIP200), and ATG101 protein [24]. Consequently, the ULK1 complex, no longer phosphorylated by mTOR but phosphorylated by AMPK, acts on the complex that regulates autophagosome nucleation [25]. It is able to activate the phosphatidylinositol 3-kinase complex (PtdIns3K) formed by Beclin1, ATG14, vacuolar protein sorting (VPS)15, VPS34, activating molecule in BECN1 regulated autophagy protein 1 (AMBRA1), and ultraviolet irradiation resistance-associated gene (UVRAG) (Figure 3A) [26]. The induction of this kinase complex generates the lipid phosphatidylinositol-3-phosphate (PI3P), which in turn recruits other proteins essential for the formation of vacuoles (such as the WD-repeat protein interacting with phosphoinositides (WIPI) protein, which is able to interact with the phospholipids) [27].

In addition to the inhibition of mTOR and activation of AMPK, the activity of the PtdIns3K complex is further regulated by the Beclin1-Bcl-2 complex [28]. When nutrients are abundant, the Beclin1 protein forms a complex with the Bcl-2 protein, which inhibits Beclin1 recruitment and function in the PtdIns3K complex; instead, when there is nutrient deficiency, the Beclin1-Bcl-2 complex is dissociated, and Beclin1 is free to act (Figure 3A) [29]. The phases of formation of the phagophore and of the autophagosome are essentially regulated by the ATG12-ATG5-ATG16 complex, necessary for there to be the lengthening of the membrane to form the phagophore (Figure 3B) [30]. A transmembrane protein system, containing ATG9, ATG2 and WIPI 1/2, is also necessary for the elongation of the phagophore structure [31]. The second conjugation complex, essential for the progress of the process, is the ATG8 protein, identified as microtubule-associated protein 1 light chain 3 (MAP1- LC3 or shortly LC3) in mammals [32]. This protein exists in its inactive form, free in the cytosol; the C-terminal end of the LC3 protein is cleaved by the ATG4 protease, thus producing a new form, called LC3-I [33]. The latter is subsequently conjugated to phosphatidylethanolamine (PE), thanks to the ATG3/ATG7 system. Lipid conjugation converts LC3-I to the LC3-II, which is exposed on the external side of the mature autophagosome [34]. Once the autophagosome is complete, it begins its journey along the microtubule towards the lysosome, with which fusion takes place [35]. The transport along the microtubule is mediated by an adaptor protein complex formed by LC3, Rab7, and FYCO1 [36]. Following the formation of autophagolysosome, the LC3-II protein is internalized, the PE residue is detached thanks to the action of lysosomal enzymes and the protein is released into the cytoplasm, with a consequent its decreased expression (Figure 3B) [37].

## 4. Selective Autophagy

Cell energy, needed to survive the unfavorable conditions caused by the absence of nutrients, derives from the non-selective degradation of damaged cellular components. In other situations, however, such as stress that damages the endoplasmic reticulum or cellular metabolism alterations, the cytoplasmic organelle to be eliminated is specifically identified by the autophagic machine and eliminated through a mechanism called “selective autophagy” [38]. There are different kinds of selective autophagy, depending on the organelles eliminated: mytophagy (for mitochondrion), ribophagy (for ribosomes), reticulophagy (for endoplasmic reticulum), lysophagy (for lysosomes), pexophagy (for peroxisomes), lipophagy (for lipid drops), glycophagy (for glycogen), aggrephagy (misfolded proteins), and xenophagy (infected pathogens) [39].

Organelles can be selectively degraded through ubiquitin-dependent or -independent mechanisms [40]. In the first case, the target organelle has a polyubiquitinated chain that directs it towards substrate-specific receptors such as p62, the neighbor of BRCA1 gene 1 (NBR1), nuclear domain 10 protein 52 (NDP52), and optineurin [41,42,43,44]. These proteins act as molecular adaptors, as they are able to recognize and bind, on one side, the polyubiquitinated chain (UB) on damaged organelles through UB-binding domain (UBD), and on the other, the LC3-II protein in the autophagosome through LC3-interacting regions (LIR) (Figure 4A). Thus, aside from its role in the autophagosome expansion, LC3-II also plays an essential role in cargo selection.

p62, a 62 kDa protein also known as sequestosome 1 (SQSTM1), is commonly localized to ubiquitinated protein and sequestered into autophagosome vacuoles [45]. It is widely expressed across tissue, such as in the nervous, endocrine, reproductive and immune systems, where it plays an important role in the pathogenesis of diseases [46]. p62, as the main selective substrate, is recruited to preexisting isolation membranes through interaction with LC3. Indeed, additional studies demonstrate that p62 localizes to the autophagosome formation site on the endoplasmic reticulum without interaction with LC3 or other ATG proteins [47]. Because of its role in delivering autophagic cargo, p62 expression is inversely correlated with autophagic degradation and can be used as a measure of autophagic flux [14].

Another cargo receptor is NBR1, which shares several domains and features with p62, as the C-terminal domain that binds polyubiquitinated chains, and as the sequence that interacts with ATG8 [48]. Recently, it has been demonstrated that NBR1 and p62 act together to selectively target polyubiquitinated protein aggregates to autophagosomes and to degrade peroxisome through pexophagy [49,50].

With regard to the ubiquitin-independent mechanism, numerous autophagic receptors on organelles directly link their cargo with the autophagosome (Figure 4B). As an example, mitophagy can involve Bcl-2/adenovirus E1B 19-kDa interacting protein 3 (BNIP3) or BNIP3-like (known as NIX), which interact directly with the LC3-II of the autophagosome [51].

Although non-selective autophagy is essential for cell viability, for physical well-being and human health, selective autophagy plays a direct role in a wide variety of diseases [52].

## 5. Autophagy Detection Methods in Mammalian Cells

Autophagy is a very complex and dynamic process that may be difficult to analyze in in vitro or in vivo models. Many researchers, who in recent years have put forward complex studies of autophagy, have provided technical and scientific contributions towards drawing up a long publication for use in the interpretation of monitoring autophagy assays [14]. The suitable techniques for in vitro autophagy study, technical measures to be followed to avoid the formation of confusing artefacts, and outlines for the guidelines for correct data interpretation are illustrated below. The main in vitro methods for monitoring autophagy are summarized in Table 2.

The preliminary observations of autophagic samples with a phase-contrast optical microscope provide a choreographic image of the phenomenon, highlighting the presence of “holes”, which are nothing but very large vacuoles present inside the cytoplasm (Figure 5).

Detailed analysis of the same sample with a transmission electron microscope shows cytoplasmic vacuoles with single and double membranes, containing cytoplasmic organelles and material to be digested and recycled (Figure 6A). Observation of samples prepared following the standard inclusion procedures allows the different phases of the autophagic vacuole formation and maturation (initiation, nucleation, phagophore formation, autophagosome, fusion of autophagosome with lysosome and autophagolysosome) to be defined [53]. If the samples are prepared using a rapid freezing technique with plasma membrane fracture (known as freeze-fracturing technique), a deep understanding of the vacuolar molecular membrane organization is obtained (Figure 6B).

Autophagosomes can be also recognize using a particular marker that identifies them. Some characteristic proteins of the autophagic pathway can be identified using fluorescent antibodies specific for a protein, and observing the samples with the fluorescence microscope. The changes observed both in LC3 protein localization and in molecular turnover, such as the conversion from LC3-I to LC3-II, can be quantified, stimulating or inhibiting the autophagy mechanism with selective drugs by western blotting technique. To clarify the autophagy dual role as cell survival or cell death mechanism, interesting results have been obtained by silencing *ATG* genes. These genes are autophagy controllers, so once the interesting gene is silenced, it is possible to evaluate how the expression of pro-autophagic or pro-apoptotic proteins varies. These results will allow us to understand whether autophagy is involved in the examined biological process for the purposes of either cell survival or cell death [54].

From this rapid description of techniques, it easy to understand that the autophagy mechanisms are very dynamic and complex, in particular when the research needs to distinguish between autophagic cell survival or cell death or between autophagy and apoptosis. To avoid assessment errors, the simultaneous use of multiple tests is recommended to be sure of the evaluation carried out.

## 6. Autophagy, Inflammation and Aging

Autophagy has been identified as main regulator of the inflammasome; a major innate immune pathway activated by exogenous stimuli, such as pathogenic microorganisms, or by endogenous mediators, such as ROS, mitochondria damage, and environmental irritants [55]. Inflammasome activation involves formation and oligomerization of a protein complex including a nucleotide oligomerization domain (NOD)-like receptor (NLR), an adaptor protein and pro-caspase-1. This activation allows cleavage and activation of caspase-1, followed by release of proinflammatory cytokines, such as interleukin (IL)-1β and IL-18, from innate immune cells [56]. In particular, when endogenous mediators induce massive inflammatory response, they can cause tissue damage and promote the onset of inflammatory diseases. Therefore, negative or positive regulation of inflammasome is essential to ensuring a good state of health.

As demonstrated by multiple studies, autophagy can negatively regulate inflammasome activation through different mechanisms:by removing damaged organelles such as mitochondria, leading to reduced release of ROS and subsequent suppression of inflammasome activation;by p62-dependent degradation of inflammasome complexes and mitochondria;by sequestering pro-IL-1β into autophagosomes for degradation. This pathway contributes to autophagy-mediated reduction in IL-1β secretion [57,58,59].

Autophagy deficiency causes inflammasome-related inflammatory diseases; genetic studies have demonstrated that polymorphisms of *ATG16L1* gene could be associated with Crohn’s disease, a chronic inflammatory disease of the intestine [60,61,62]. Unfortunately, the pathophysiology of Crohn’s disease is still under investigation; in fact, it has been shown by some authors [63,64] that other genes and predisposing environmental factors can influence the onset of the disease in patients [65,66]. Therefore, some caution must be taken in order to distinguish the real role of autophagy in inflammatory bowel diseases.

Overall, these data suggest that inflammasome and autophagy mutually regulate each other, favoring the balance between inflammatory response to defend itself from the host and prevention of excessive inflammatory response that can induce tissue damage and inflammatory disease [67]. 

Recent studies have shown that the impaired autophagy activity that characterizes aging is due to accumulation of dysfunctional mitochondria, ROS and NLRP3 inflammasome activation in macrophages [68]. These factors predispose the cells to greater risk towards aging diseases, such as atherosclerosis and type 2 diabetes. In particular, the activities of three signaling pathways related to aging have been identified: the lower activity of insulin/insulin-like growth factor 1 (IGF-1) signaling, mTOR, and Sirtuin-1 (Sirt-1) network. For age-related diseases, it is advisable to follow a diet rich in foods containing anti-aging natural compounds as resveratrol, catechins, epigallocatechin-3-gallate, luteoloside or propolis [69]. Recent studies have demonstrated that resveratrol, a natural polyphenol of red wine, activated NAD^+^-dependent deacetylases such as Sirt-1, which is a promising therapeutic target for age-related diseases [70]. Moreover, this natural product was able to stimulate autophagy activation by suppressing NLRP3 activation [71].

## 7. Role of Autophagy in Metabolic Syndrome Diseases

The role of autophagy in metabolic syndrome is particularly interesting. Metabolic syndrome, as defined by the NHS, is a multifactorial disease, especially in industrialized countries, and is characterized by at least three of these risk factors: high blood pressure (above 85/130 mmHg);high triglycerides (above 150 mg/dL);low levels of high-density lipoprotein (HDL) cholesterol (lower than 40/50 mg/dL);high blood sugar (above 100 mg/dL);visceral distribution of body fat [72].

Simultaneous presence of these factors exposes the patient to the potential danger of developing type 2 diabetes, obesity and consequent heart disease. The onset of this pathology is very often favored by family predisposition and bad lifestyle, characterized by an incorrect and unbalanced diet and poor physical exercise. To understand the role of autophagy in the metabolic syndrome, one must dwell on the physiological defect that defines type 2 diabetes. Type 2 diabetes is characterized on the one hand by deficiency in insulin secretion and on the other hand by insulin resistance (the insulin produced does not satisfactorily act on the target organs). The combined action of these two processes is responsible for hyperglycemia, typical of type 2 diabetes (also called diabetes mellitus). The increase in blood glucose levels determines, at the systemic level, in the cells of the target organs, the following dysfunctions: reactive oxygen species increasing and antioxidant systems decreasing (oxidative stress activation); mitochondrial dysfunctions; and activation of the inflammatory process, leading to nephropathy, foot ulcers, stroke, neuropathy, retinopathy and heart disease (Figure 7).

The progression of type 2 diabetes is associated with autophagy through insulin secretion deficiency and the development of insulin resistance [73]. Autophagy has a protective role against pancreatic beta cells, preserving their structure and function and defending them from apoptotic cell death. In addition, selective autophagic mechanisms, such as lypophagy, aggrephagy, mitophagy and glycophagy, protect the insulin target organs (liver, adipose tissue, skeletal muscle, kidneys) from oxidative stress damage derived by hyperglycemia. Hence, multiple autophagy mechanism alterations modify the metabolism and the cellular balance, favoring the establishment of pathological situations such as diabetes, obesity and the cardiovascular complications [74]. 

The protective effect of autophagy on the metabolically active organs has been scientifically proven through animal studies. In the liver, for example, autophagy ensures the proper function of hepatocytes and regularity in the glycogen and triglycerides metabolism. Animal model studies in which hepatocytes underwent the deletion of the *ATG7* gene show an accumulation of altered and deformed mitochondria, of lipid droplets, and an increased number of ubiquitinated protein aggregates [75]. At the level of the pancreas, autophagy ensures beta cell structure, mass and functionality maintenance, and reduces the stress levels produced by the endoplasmic reticulum alterations [76]. In adipose tissue, autophagy ensures the differentiation of adipocytes: the removal of *ATG5* and *ATG7* genes in adipocytes causes abnormal accumulation of lipids, and abnormal transformation of white into brown adipose tissue [77]. Finally, in skeletal muscle, autophagy ensures greater tolerance to glucose and good maintenance of the mass itself. *ATG5* and *ATG7* gene deletions in the skeletal muscle cause the mice to lose muscle mass and fat mass, and also to suffer damage at the level of adipose tissue (conversion of white adipose tissue to brown and greater beta oxidation) [78]. A recent review summarizes studies related to the role of multifunctional p62 in metabolic diseases, implying that p62 might be a potential target. The p62 protein protects pancreatic beta cells against apoptosis and reduces insulin resistance; it controls the protein quality of cardiomyocytes by autophagy activation and plays a protective role in attenuating the development of diabetic nephropathy by inhibiting epidermal growth factor receptor (EGFR) activation [79]. 

Considering the central role of autophagy in metabolic syndrome, the best therapeutic strategy to fight this disease is the use of autophagy as a target through caloric restriction and intermittent fasting, physical exercise and, if necessary, pharmacological treatment (Figure 8) [80].

Fasting and caloric restriction have beneficial effects on human health [81], since they are able to restore the physiological levels of autophagy through the activation of Sirt-1 and AMPK (Figure 8A) [82,83]. Sirt-1 is a NAD^+^-dependent deacetylase involved in glucose metabolism and insulin secretion; this protein is inhibited in cells that have a high resistance to insulin [84]. It functions as a metabolic sensor that detects the increase in NAD^+^ concentrations resulting from enhanced NADH oxidation. Once activated by caloric restriction, Sirt-1 can deacetylate essential autophagic modulators, such as ATG5 and ATG7, stimulating its autophagic effects [85]. Moreover, fasting and caloric restriction are able to decrease vascular dysfunction and cardiovascular risk associated with metabolic diseases through modulation of inflammation cytokines and oxidative stress [86]. In particular, fasting increases transcription of stress-induced proteins, such as heat shock protein 70 (HSP70), which has anti-inflammatory and anti-apoptotic properties. In the skeletal muscles of diabetic patients, levels of HSP70 decrease, which is related to insulin resistance; so elevations of HSP70 level by fasting mitigate insulin resistance and consequent vascular dysfunction [87].

The beneficial effects of physical exercise in people with type 2 diabetes and cardiovascular complications are well documented. Physical exercise activates a cascade of events that leads to a slight increase in oxidative stress, energy imbalance, intracellular calcium levels and the remodeling of proteins. All these events are able to activate the autophagy survival mechanism, which facilitates the mitochondria and protein turnover and metabolism modification. These adaptive responses generally lead to lipid and glucose homeostasis optimization and resistance performance improvement [88]. In particular, in skeletal muscle cells, physical activity works by destroying the Beclin1–Bcl-2 complex, which is essential for the start-up phase of autophagy (Figure 8B) [89].

Physical exercise plays a beneficial role in obese individuals. In the skeletal cells of obese subjects, defects in the mitophagy may lead to oxidative stress, mitochondria accumulation and malfunction and alteration of the equilibrium between fission and fusion mitochondrial processes (fundamental balance for maintaining a good state of mitochondria health). Although molecular and cellular mechanisms are not clearly elucidated, physical exercise intervenes by restoring the physiological levels of mitophagy and the fusion and fission processes reducing apoptotic events in obese skeletal muscle [90]. 

With regard to pharmacological interventions, diabetic patients are treated with metformin, a hypoglycemic drug with many side effects. Metformin, acting on the AMPK protein, is able to activate both mytophagy and lipophagy and to block the activation of the inflammasome [91]. Considering the side effects of metformin and the onset of drug resistance, in light of the new knowledge related to the molecular pathways that regulate autophagy, the search for natural substances with hypoglycemic effects is constantly evolving [92]. Recent studies demonstrated that the extracts from *Moringa oleifera* plant, rich in flavonoids, isothiocyanates and polyphenols, exhibit multiple functions, including anti-inflammatory, hypoglycemic and blood lipid-reducing function [93]. The molecular mechanisms responsible of these multifunctions are the inactivation of NF-kB and PI3K/AKT pathways and modulating protective action on mitochondrial respiratory chain mediated by Sirt-1 (Figure 8C) [94].

## 8. Autophagy and Neurodegenerative Disorders

The aggregation of misfolded proteins and some neuronal population losses are typical of the expression of pathological neurodegenerative diseases such as Alzheimer’s disease, Parkinson’s disease, Huntington’s disease and amyotrophic lateral sclerosis. Autophagy has been reported to be involved in the occurrence of neurodegenerative disorders, being the main intracellular system for degrading damaged organelles and aggregated proteins. In neurodegenerative diseases, an alteration of the maturation mechanism of the autophagosome in the autophagolysosome has been found.

It has been shown that *ATG5* and *ATG7* gene depletion causes neurodegeneration in the murine central nervous system [95].

Moreover, autophagy plays an important role in the degradation of different proteins correlated with degenerative diseases, such as mutated α-synuclein in Parkinson’s disease, mutant huntingtin in Huntington’s disease, and the mutant TAR DNA-binding protein 43 (TPD-43) in amyotrophic lateral sclerosis. 

In Alzheimer’s disease, the presence of extracellular amyloid-β plaques and intracellular neurofibrillary tangles, composed of hyperphosphorylated tau proteins aggregates, has been revealed [96]. In the healthy brain, the autophagosome vesicles are not very visible; instead, in the Alzheimer’s disease brain, numerous autophagosomes are noticeable. Accumulation of autophagy vacuoles arises from impaired clearance rather than autophagy induction, suggesting the late stages of autophagy modulation as a possible therapeutic strategy for Alzheimer’s disease. The transmembrane protein involved in autophagosome and lysosome fusion is Presenilin 1 (PS1). This mutated protein is considered to play a key role in the induction of for Alzheimer’s disease; in fact, the loss of PS1 Ser367 phosphorylation prevents the fusion of two vacuoles [97]. Further studies show that PS1 preserves the Ca^2+^ homeostasis by regulating the lysosome acidification [98].

The reduction of acidification alters the lysosome functionality, preventing its fusion with the autophagosome. This alteration leads to neuron dystrophy similar to that for Alzheimer’s disease; from these observations, we understand how this organelle and its correct functionality are important for the removal of the aggregated proteins.

Other proteins are correlated with Alzheimer’s disease, such as the clathrin group binding the phosphatidylinositol protein, which is involved in vesicular trafficking, which regulates endocytosis and prevents the aggregation of tau proteins and Beclin1. Beclin1 has a key role in the formation of the autophagosome; Caspase 3, an important apoptosis activation component, is able to separate the Beclin1 protein and arrest the progression of the autophagy process.

It is known that some genetic and biochemical defects can initiate Alzheimer’s disease, inducing oxidative stress and chronic inflammation. Regarding this, the increase of antioxidant systems and the corresponding enzyme levels seem to induce beneficial effects on neurodegenerative disease in animal models. A nuclear factor involved in the Alzheimer’s disease pathology and activated in response to the oxidative stress is the nuclear factor erythroid derived 2 like 2 (Nrf2) [99]. This factor is hypothesized to activate autophagy and eliminate aggregated tau proteins, through the autophagic receptor NDP52 [100,101].

Parkinson’s disease is a widespread neurodegenerative disease and is characterized by a severe loss of dopaminergic neurons in the substantia Nigra pars compacta, followed by polyubiquinate and α-synuclein protein inclusions called Lewy bodies. These bodies appear to be characteristic of Parkinson’s disease, but also of other neurological disorders [102]. 

The role of autophagy in Parkinson’s disease has been demonstrated by the presence in neurons of lysosomal and autophagosomes alterations; to support this evidence, when the lysosome is functionally altered, the amount of α-synuclein is elevated, indicating an alteration of the autophagy pathway. It has been shown that gene GBA autosomal recessive mutation, which encodes for lysosomal hydrolase, induces alterations in the autophagosome-lysosomal pathway and aggregation of α-synuclein [103]. The EB transcription factor (TFEB) has been identified as the factor that positively regulates genes related to formation of autophagosomes and the lysosome fusion, increasing the clearance of lysosomal exocytosis [104]. Recently, it has been shown that its overexpression can reduce the lysosome damage and thus improve the neurological disorders related with α-synuclein. 

The common causes of autosomal dominant forms of Parkinson’s disease are due to mutations in leucine-rich repeat Kinase 2 (LRRK2) and in VPS35 D620N. In the case of LRRK2, it has been shown that its up-regulation alters the autophagy flow, while the mutation of VPS35 D620N changes the trafficking of the ATG9 protein through the alteration of the WASH complex. The autosomal recessive forms of Parkinson’s disease are caused by mutations of Parkin RBR E3 ubiquitin protein ligase (PARK2) and PTEN-induced putative kinase 1 (PINK1), which regulates degradation of damaged mitochondria (mitophagy). PINK1 acts as an upstream factor, accumulating specifically on depolarized mitochondria, while PARK2 catalyzes the transfer of ubiquitin to mitochondrial substrates [105,106]. 

Huntington’s disease is a neurodegenerative disease caused by GAG trinucleotide repeat in the gene encoding the huntingtin protein (HTT). Huntingtin has an important role in the trafficking of autophagy proteins. HTT depletion causes an abnormal accumulation of autophagosomes with engulfed mitochondria. In Huntington’s disease, an *ATG7* gene polymorphism, an alteration in both Beclin1 and in the selective substrate p62/SQSTM1 of autophagy, can be observed. Dysregulation of these systems are also very important in the early onset of Huntington’s disease. On the contrary, non-mutant HTT can bind p62, interact with ULK1 and induce autophagy [107].

Amyotrophic lateral sclerosis is a neuromuscular disease in which there is a progressive degeneration of spinal cord and brain motor neurons up to atrophy and muscular paralysis. To date, around 30 genes have been implicated, including those involved in autophagy and mitophagy. In particular, an alteration of the clearance of protein aggregates and damaged mitochondria is observed. The genes that have been associated with amyotrophic lateral sclerosis encode proteins that bind RNA such as TDP43 and FUS, thus indicating that ribostasis alterations may favor the disease. At the onset of the disease, alterations of the protease, mitochondrial function, cytoskeleton integrity and intracellular traffic are also involved. The amyotrophic lateral sclerosis genes involved in autophagy are SQSTM1, TBK1 and OPTN. Instead, the genes regulating vesicular traffic such as C9ORF72, VCP, CHMP2B, VAPB, ALS2, DCTN1 may also be involved in the autophagy mechanism, both directly and indirectly. In amyotrophic lateral sclerosis, motor neurons of patients, accumulation of protein aggregates and swollen or dystrophic mitochondria are often found. There is currently no cure for this disease [108].

## 9. Pro-Survival Autophagy in Cancer

The role of autophagy in cancer is controversial and complex, and depends on the cancer type, stage, and genetic context [10]. During the early phase of cancer development, autophagy protects cells from ROS-induced damage to DNA and proteins, slowing down the transformation of normal cells into tumor cells [109]. It has been confirmed that mice with monoallelic deletion of Beclin1 have an increased susceptibility to spontaneous tumor development [110]. 

During the late stages, such as promotion, progression and metastasis, autophagy has a pro-tumoral effect, because it eliminates ROS-induced metabolic stress products and supplies nutrients required for cancer cell survival [111]. 

The role of autophagy is particularly interesting in cancer stem cells (CSC), which are characterized by elevated levels of autophagy compared to more differentiated cancer cell populations, as observed in multiple cancer types, including urinary bladder and breast cancer [112,113]. These high levels of autophagy are essential to maintaining CSC properties such as dormancy. 

Autophagy plays a regulatory role during cancer cell adaptation to hypoxic stress in poorly oxygenated regions of solid tumors or in the area of bone marrow after acute myeloid leukemia infiltration [114].

Moreover, the onset of autophagy allows cancer cells to protect themselves against chemotherapeutic toxin action, blocking the apoptotic effect induced by drug treatment (autophagy-mediated resistance) [115]. For example, cis platinum-resistant ovarian cancer cells are characterized by enhanced levels of autophagic flux [116].

Finally, there are drugs able to induce autophagic cell death that are used to achieve a greater therapeutic result on apoptosis-resistant tumor cells [117,118].

The co-existence of various cellular and molecular processes within a tumor mass has led the research towards identification of key molecules in the autophagy/apoptosis switch (Beclin1, Caspase, p53, PI3K/AKT/mTOR, and p62), which is useful for planning anti-cancer target therapies [119,120].

Preclinical studies have highlighted the importance of suppressing autophagy in therapeutic strategies. KRAS is a gene that controls cell proliferation; if it is mutated, the cells continue to proliferate by developing into cancer. BRAF is a gene that codes for a protein known as serine/ threonine-protein kinase B-Raf. This protein is involved in the activation of cellular signals towards cell proliferation; in many human tumors, this gene is defective [121,122]. In human cell lines that exhibit such mutations, high levels of autophagy were observed. KRAS and Tp53 status is also very important; indeed, in KRAS-mutant mice lacking *ATG5* or *ATG7* genes, pre-malignant lesions in the pancreas were revealed, while in the KRAS and Tp53-deficient mice, the loss of *ATG5* or *ATG7* genes induced the acquisition of malignancy and the development of adenocarcinomas [123]. These results suggested that autophagy inhibition in BRAF or KRAS mutation in tumors could be a potential anticancer target.

In recent years, many compounds able to induce or inhibit autophagy have been identified for pharmacological intervention, but their use must be widely supported by preclinical data [124]. Torin1, a potent and selective inhibitor of mTOR, and Rapamycin, an mTORC1 inhibitor, are able to stimulate autophagy [125]. Autophagy is inhibited at multiple levels by 3-methyladenine, LY294002, Wortmannin (PI3K specific inhibitors), SBI-0206965 (ULK1 inhibitor), Spautin1 (inhibitor of UB-specific peptidase), Vinblastine (blocks the fusion between lysosome and autophagosome), E64d and Pestatin A (lysosomal protease inhibitors), Bafilomycin A1 (ATPase pump inhibitor), and Chlorochine (a lysosomotropic compounds that interferes with autophagosome degradation) (Figure 9) [124,126].

Hydroxychloroquine, the chloroquine antimalarial derivative drug, has been tried out in clinical trials (Table 3); it is capable of blocking the lysosomal enzyme degradative action within the autophagolysosome. Hydroxychloroquine is used instead of chloroquine, because its autophagy inhibitory action is less toxic. Many clinical studies have shown its efficacy when administered alone or in combination with other chemotherapeutics, and other studies have proved it to be effective against different solid tumors, such as glioblastoma, astrocytoma, lung and pancreatic cancer [10].

Moreover, many natural products from living organisms have biological and pharmacological properties that modulate autophagy; these potential applications may be studied on in vitro models and translated into clinical application [127].

## 10. Autophagy and Autoimmune Diseases

Autoimmune diseases, such as systemic lupus erythematosus (SLE), are characterized by significant changes in the prognosis and outcome of the treatments. They involve the entire immune system, including B and T cells of the adaptive arm, and dendritic cells, macrophages, and neutrophils of the innate arm, with the consequent production of antinuclear bodies. The metabolic pathways are important regulators of the immune system differentiation and activation, so if they are altered, they can lead to the development of autoimmune diseases [128]. Metabolic control of immune system differentiation starts on hematopoietic stem cells, which activate mTOR-dependent autophagy [129]. mTOR is a nutrient-sensing kinase that, linking metabolic signals to genetic patterns, controls cell growth and differentiation. In particular, mTOR forms two complexes: mTORC1, which drives Th1 and Th17 development, and mTORC2, which mediates Th2 development; both complexes restrict regulatory T cells (Treg) development [130]. During metabolic stress, activation of endosomal traffic by Rab4A promotes autophagy through CD4 and CD3 surface receptors and the dynamin-related protein 1 (Drp1) mitochondrial fission initiator. Lysosomal degradation of Drp1 decreases mitophagy, with consequent accumulation of large elongated mitochondria and ROS-generation in lupus T cells [131]. ROS activates mTORC1, causing proinflammatory necrotic death and IL-4 and IL-17 secretion by DN Tcells and depletion of CD8 memory T cells and Treg cells [132,133]. In the liver, this activation precedes the production of antinuclear antibodies and disease onset in lupus-prone mice [134]. These data show a pathogenic role for Rab4-mediated Drp1 depletion and identify it as potential target for SLE treatment [131]. While autophagy has been amply examined in SLE T cells, autophagy in B cells has been less studied. Clarke and co-authors demonstrate an enhanced autophagy in murine and human lupus B cells, and that it is required for plasmablast development [135]. They demonstrated the important role of autophagy in plasmablast differentiation, because B cells isolated by *ATG7* deficiency mice after in vitro stimulation failed to effectively differentiate into plasma cells. Similarly, stimulated human B cells, after autophagy inhibition, did not differentiate in plasmablasts. Hence, they found, before disease onset, in an early stage of B cells of SLE mouse model, an activation of autophagy that increased with age [135].

The imbalance between peripheral Th17 and Treg cells is critical to SLE pathogenesis and other autoimmune diseases [136]. Th17 cells, producing IL-17A, IL-17F and IL-22, promote the autoimmune response and favor autoantibody production by B cells [137]. Treg cells control the immune homeostasis, releasing IL-10 and TGF-β. The most important Treg-specific transcription factor is forkhead box protein 3 (Foxp3); the expression decrease of Foxp3 in Treg cells is responsible for Treg dysfunction in SLE [138]. Considering the pivotal role of autophagy in the metabolic checkpoints and in the autoimmunity control, specifically in the SLE pathogenesis, recent studies have addressed autophagy-targeted therapies.

Some studies have reported that HCQ is a mainstay of SLE and other immune diseases, such as rheumatoid arthritis treatment [139]. Both Th17 and Treg cells in SLE exhibit autophagy activation, and by blocking the autophagy pathway with CQ, the Th17/Treg response could be rebalanced. Thus, the inhibition of autophagy reduces IL-17 secretion by reversing the Th17 cells over activity observed in SLE and increases Foxp3 expression in Treg cells, preserving Treg cell functionality. Thus, the immune balance is restored, lowering the serum levels of inflammatory cytokines and autoantibodies [140].

Patients with SLE have Treg dysfunction due to mTOR activation. Rapamycin, acting on mTOR, inhibits Tcell proliferation and has been developed as medication under the generic designation of sirolimus [141]. In vivo study demonstrated that sirolimus abrogated autoimmunity in lupus-prone mice [142]. A recent prospective study demonstrated the efficacy and safety of sirolimus in SLE patients with severe and persistent disease [143]. 

Considering that SLE can be associated with an increased incidence of malignancy, mainly lymphomas, the treatment with immunosuppressive drugs could also contribute to the development of secondary cancers [144]. Emerging data suggest that rapamycin is an effective antineoplastic agent in leukemia and lymphoma, and hence could be used as an immunosuppressive agent for treatment of SLE patients [145]. However, it is not always clear whether pharmacological modulation of autophagy has positive or negative effects; thus, more detailed studies must be performed to develop a specific target therapy. 

## 11. Conclusions

Autophagy is a highly conserved degradation process throughout evolution. Many Nobel Prizes have been assigned to researchers who have directed their attention to the study and understanding of this phenomenon: in 1974, Dr. Christian de Duve described autophagy as a process of “self-eating”, and in 2016 Dr. Yoshinori Ohsumi was recognised for the remarkable contribution towards identification of essential genes for autophagy [146,147].

Numerous studies have shown that it plays a fundamental role in maintaining cellular energetic and physiological balance; therefore, the alteration of this phenomenon is among the causes of increasingly frequent illnesses. Many pharmacological treatments and appropriate lifestyles are aimed at restoring the autophagy physiological levels, suggesting that this process is a potential and promising target of public and health interest.

## Figures and Tables

**Figure 1 ijms-20-00725-f001:**
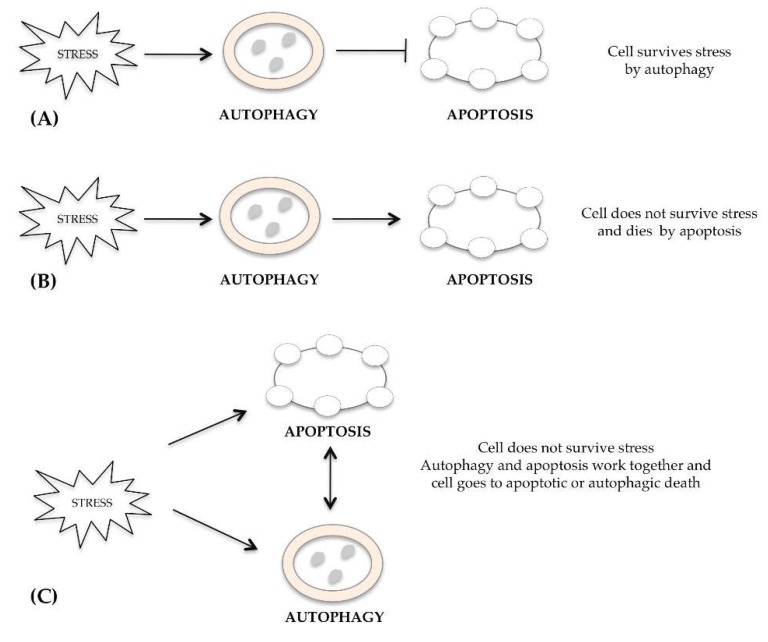
Cross-talk between autophagy and apoptosis. (**A**) After stress signal, cells survive by activating the autophagic pathway, which blocks death by apoptosis; (**B**) Alternatively, the autophagic mechanism fails and cells dies by apoptosis; (**C**) Damage activates both autophagy and apoptosis, which work together; cell does not survive, but dies.

**Figure 2 ijms-20-00725-f002:**
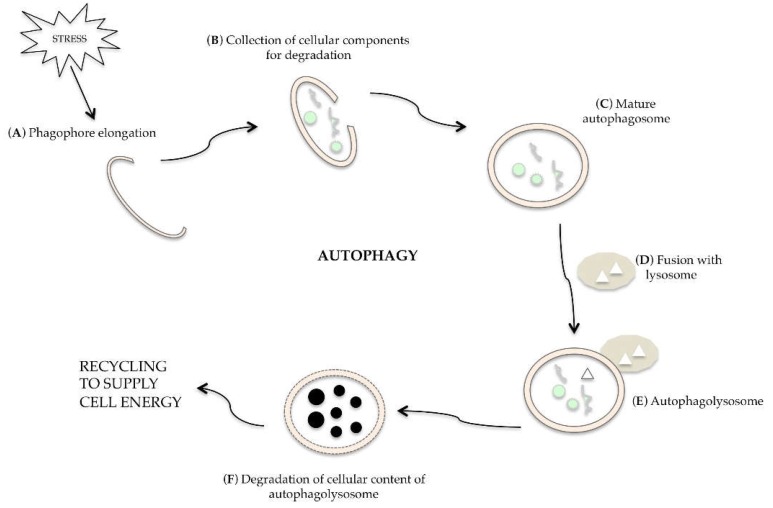
Description of cell autophagic phases: upon various stresses, a small double layer membrane, called phagophore, extends inside the cells (**A**) to collect cellular material to degrade (**B**) and to form autophagosome (**C**). After fusion with lysosome (**D**), the autophagolysosome is formed (**E**); thanks to the action of hydrolytic enzymes, cellular components are degraded (**F**) and recycled to supply cell energy.

**Figure 3 ijms-20-00725-f003:**
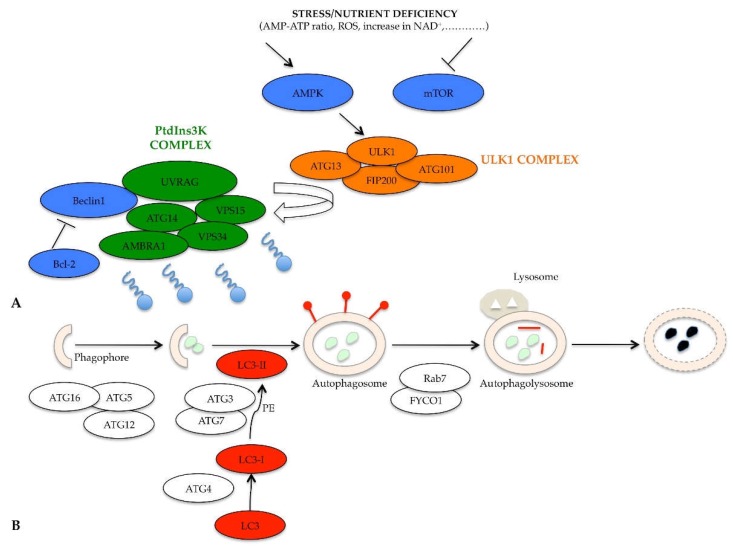
Schematic representation of molecular pathway of autophagy. (**A**) Upstream activation characterized by mTOR inhibition, AMPK activation, ULK1 complex and PtdIns3K activation; (**B**) Regulation of phagophore elongation by ATG12-ATG5-ATG16 complex, maturation of autophagosome by ATG8/LC3 complex, end of the process mediated by Rab7, FYCO1 transport proteins.

**Figure 4 ijms-20-00725-f004:**
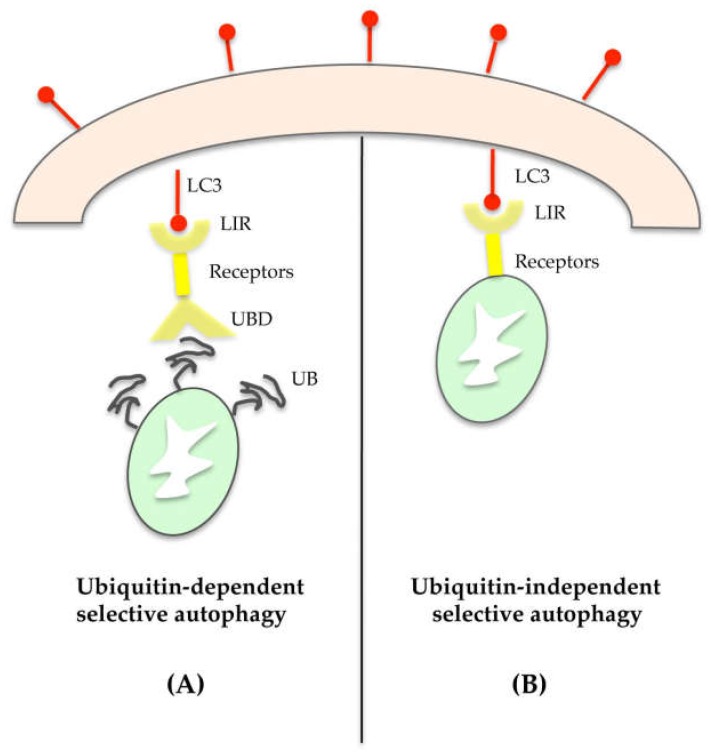
Selective autophagy can be ubiquitin-dependent (**A**) or ubiquitin-independent (**B**). In the ubiquitin-dependent mechanism (**A**), the target organelle interacts with the receptor (as p62, NBR1, NDP52, optineurin) through a polyubiquitinated chain (UB). The receptor protein binds on the side Ub chain of the damaged organelle through the UB-binding domain (UBD), and on the other, the LC3-II protein of the autophagosome through LC3-interacting regions (LIR). In the ubiquitin-independent mechanism (**B**), receptors localized on the damaged organelle (as BNIP3, NIX) directly link cargo with LC3 on the autophagosome.

**Figure 5 ijms-20-00725-f005:**
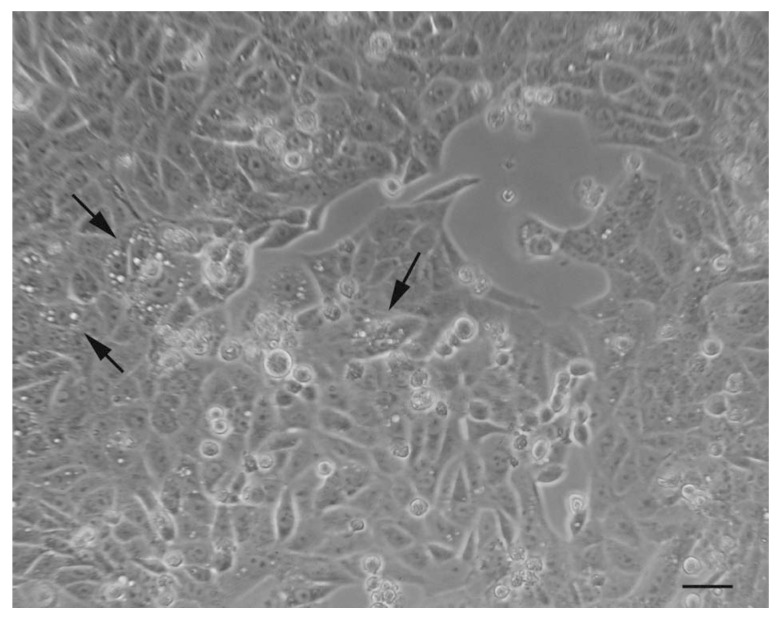
Optical microscopic observation of autophagic tumor cells after treatment with natural compound. As indicated by arrows, there are visible holes inside the cytoplasm. Scale bar: 20 μm.

**Figure 6 ijms-20-00725-f006:**
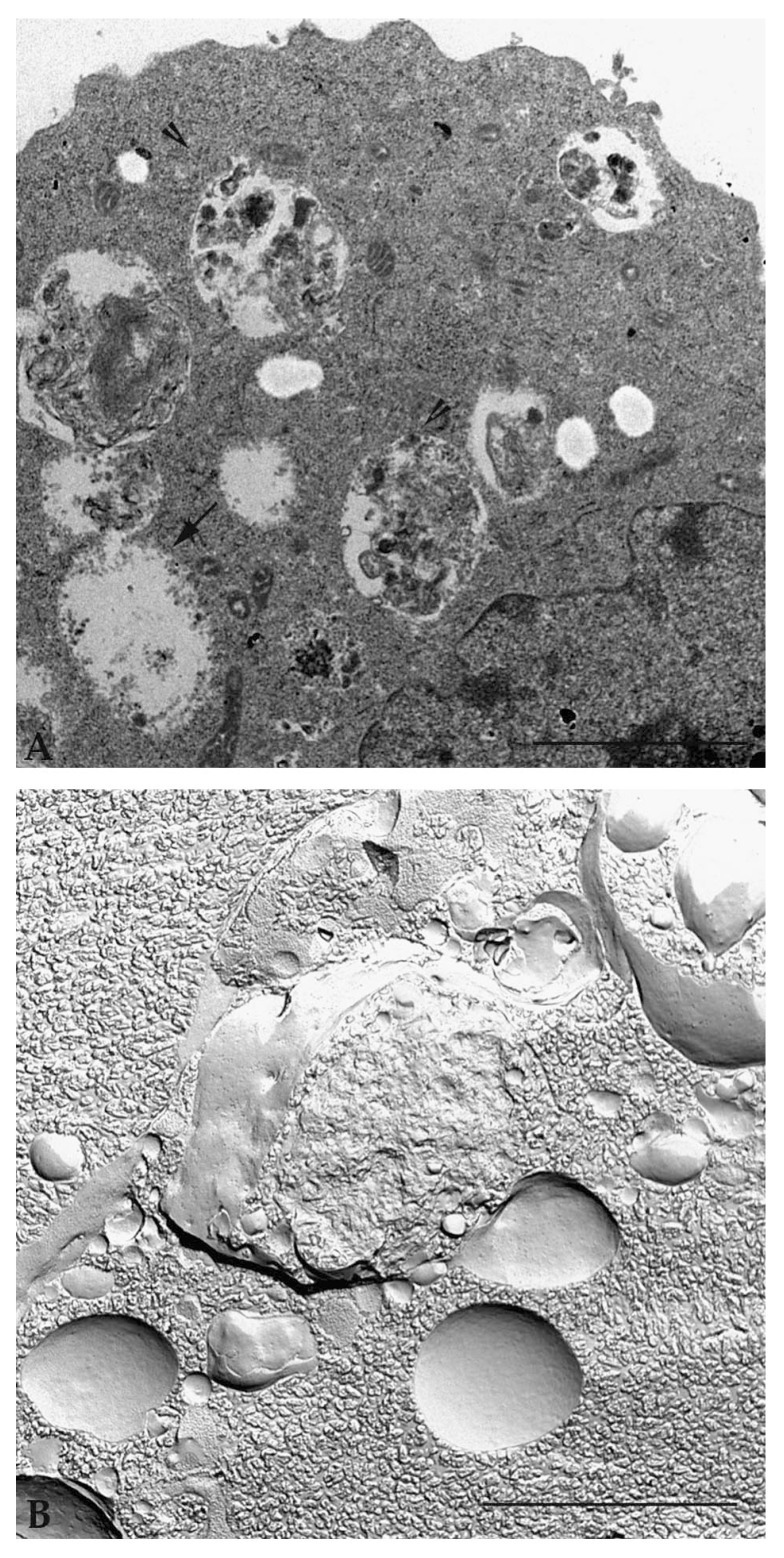
Transmission electron microscopic observation of autophagic tumor cells, after treatment with natural compound. (**A**) Observation of ultrathin sections shows vacuoles at different stages of maturation: empty vacuoles (arrow), vacuoles with digested and strongly electrondensated materials (arrowhead); (**B**) Observation of sample prepared by freezing technique shows vacuoles with multilayer membrane. Scale bar: 2 μm.

**Figure 7 ijms-20-00725-f007:**
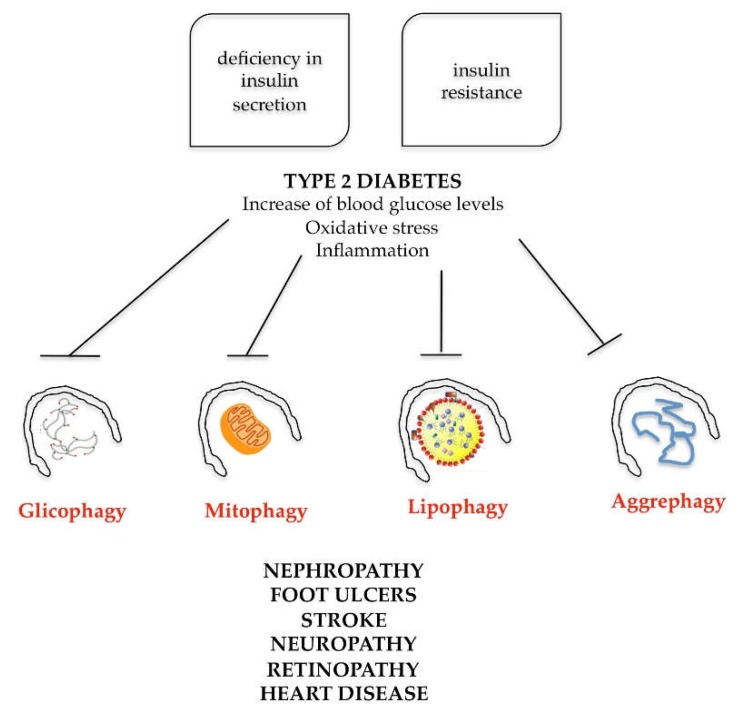
Schematic representation of autophagy involvement in metabolic syndrome. Type 2 diabetes, due to deficiency in insulin secretion or insulin resistance, increases blood glucose levels, oxidative stress and inflammation. These cellular events are able to inhibit different types of selective autophagy, such as glicophagy, mitophagy, lipophagy or aggrephagy with consequent systemic effects.

**Figure 8 ijms-20-00725-f008:**
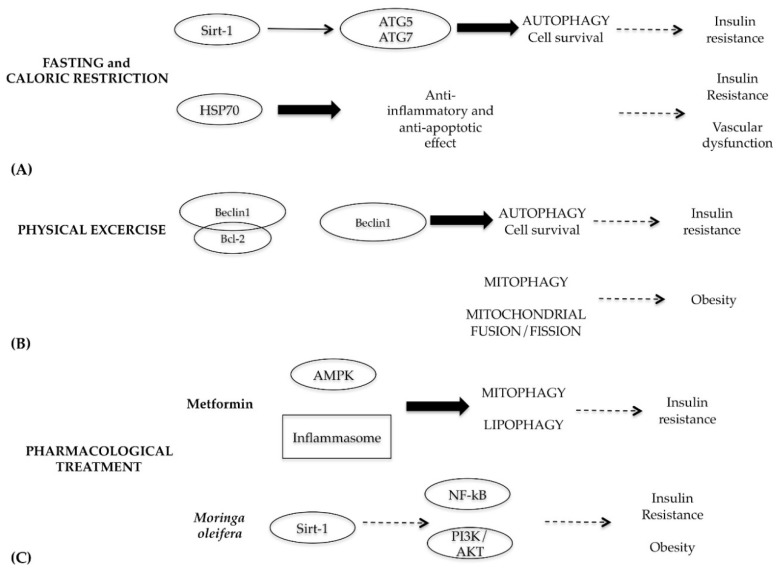
Schematic representation of autophagy modulation by fasting and caloric restriction, physical exercise or pharmacological treatment. (**A**) Fasting and caloric restriction acts on Sirt-1 and HSP70 proteins with autophagic and anti-apoptotic effect, respectively; (**B**) Physical exercise induces autophagy, disrupting the Beclin1 and Bcl-2 complex and balancing mitochondrial fusion/fission; (**C**) Treatments with Metformin or natural extract by *Moringa oleifera* restore mitophagy and lipophagy.

**Figure 9 ijms-20-00725-f009:**
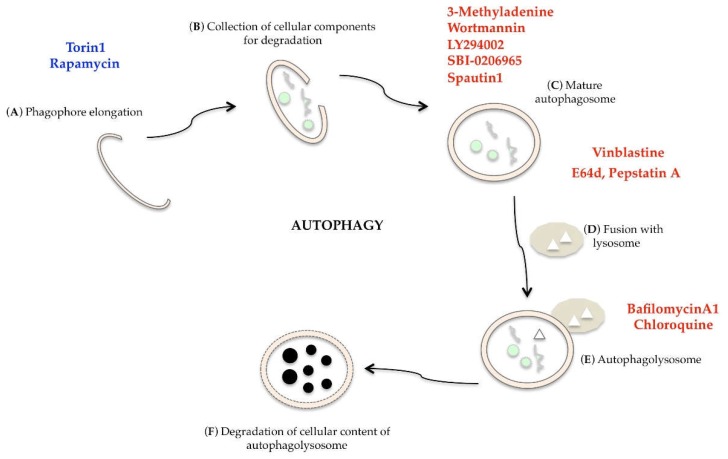
Modulation of autophagic single phases: autophagic inducers are in blue (Torin1 and Rapamycin), autophagic inhibitors are in red (3-Methyladenine, Wortmannin, LY294002, SBI-0206965, Spautin1, Vinblastine, E64d, PestatinA, BafilomycinA1, Chloroquine).

**Table 1 ijms-20-00725-t001:** The main morphological, biochemical and molecular characteristics of apoptosis, necrosis and autophagy.

Types	Morphological Features	Biochemical Features	Core Regulators
**Apoptosis**	Plasma membrane blebbing Rounding-up of the cell Reduction of cellular volume Reduction of nuclear volume Nuclear fragmentation Chromatin condensation	Phosphatidilserine exposure Activation of caspases Dym dissipation DNA fragmentation	Positive: p53 Bax Bak Other pro-apoptotic Bcl-2 proteins Negative: Bcl-2, Bcl-XL Other anti-apoptotic Bcl-2 proteins
**Necrosis**	Rupture of plasma membrane Cytoplasmic swelling Moderate chromatin condensation	Drop in ATP levels Release of inflammatory cytokines	Positive: RIP1 RIP3
**Autophagy**	Accumulation of double membrane vacuoles Lack of chromatin condensation	LC3-I to LC3-II conversion Substrate (e.g., p62) degradation	Positive: ATG5 ATG7 Beclin1 Other ATG proteins

**Table 2 ijms-20-00725-t002:** The main in vitro methods for monitoring autophagy.

Method	Description
Optical and electron microscopy	Display vacuoles inside the cytoplasm, their content, their stage of maturation, and the turnover of autophagic compartments
GFP-LC3 fluorescence microscopy	Monitor vacuolar/lysosomal localization
LC3 western blotting	Monitor autophagic flux with or without autophagic modulators
Flow cytometry	Autphagosome quantification by fluorescent probes
Western blot	p62 and related LC3 binding protein turnover
Kinase assays or western blotting	mTOR, AMPK and ULK1 kinase activity
*ATG* genes silencing	Allows to identify ATG target proteins involved in the phenomenon

**Table 3 ijms-20-00725-t003:** List of some clinical trials of hydroxychloroquine.

Combination of Treatment	Type of Tumor	Trial on *ClinicalTrials.gov*
Sirolimus or vorinostat + HCQ	Advanced solid tumors	NCT01266057
Single HCQ	Glioblastoma and astrocytoma	NCT02432417
HCQ + Vorinostat	Malignant solid neoplasm	NCT01023737
Cisplatin, etoposide + HCQ	Stage 4 small cell lung cancer	NCT00969306
HCQ + Abraxane + Gemcitabine	Pancreatic adenocarcinoma	NCT01978184
Sorafenib + HCQ	Refractory or relapsed solid tumors	NCT01634893

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
