# Peer review of "Targeting Autophagy to Overcome Human Diseases"

_ijms, 2019, doi:10.3390/ijms20030725_

Round 1
Reviewer 1 Report
The review is very interesting , in particular the overview on autophagy involved pathologies. It also opens new scenarios for specific experimental models aimed at autophagy characterization in vitro.
Author Response
I thank the reviewer for showing great interest and having appreciated this review article.
Reviewer 2 Report
The authors have done a decent job of covering a large amount of information in a condensed setting. However, I have several criticisms (see below).
Criticisms
The authors have mentioned that loss of a Beclin1 allele can lead to cancer and that polymorphisms of ATG16L1 gene are associated with Crohn’s disease. This are both true, but the authors should also indicate that in both cases is far from certain that these effects are the result of their effects on autophagy itself, as multiple publications have new questioned these relationships, and suggested that that it is from autophagy-independent functions of these molecules and/or passenger deletions that may actually be relevant. The authors should inserts some cautionary sentences that describe the results of these studies.
A second criticism is in the authors treatment of what is termed programmed cell death type II, which in our experience is quite rare and for the most part is referenced by papers that do not actually examine whether autophagy is actually a cause of death versus just accompanies death (since autophagy is activated in response to stressors, this is not surprising). For instance, many “papers claiming a causative role of autophagy in cell death fail to provide adequate evidence”. (See information in Cell Death Section in Guidelines for the use and interpretation of assays for monitoring autophagy, as cited by the authors.) There are legitimate studies where autophagy seems to actually be causative of cell death in physiological systems, such as in the Drosophila Salivary glands, for instance, but in our view the limited nature of genuine proof of “autophagic cell death’ is a cause for concern when review papers treat it as a well -accepted phenomenon without acknowledging the prevalence of misleading and incorrect studies. The problem deepens when further investigators continue to do flawed studies and then cite these publications as evidence for a previous foundation for their papers. It is therefore our recommendation that cautionary sentences, such as are found in Guidelines, be inserted in this section.
Lastly, the model for figure 4 is incorrectly illustrated as interaction between LIR containing adaptors is shown as the sole factor for recruiting these proteins to the autophagosome. While this interaction is certainly involved, adaptors, such as p62, can be recruited in the absence of LC3 (See J. Cell Biol. Vol. 192 No. 1 17–27), and thus the interaction is probably a stabilizing interaction, rather than a direct recruitment.
The manuscript would be potentially acceptable after addressing these criticisms.
Author Response
The authors have done a decent job of covering a large amount of information in a condensed setting. However, I have several criticisms (see below).
Response: I thank the reviewer for showing interest in this review, giving us the opportunity to further improve the quality of our work. Here below answers and integration requested.
Point 1: The authors have mentioned that loss of a Beclin1 allele can lead to cancer and that polymorphisms of ATG16L1 gene are associated with Crohn’s disease. This are both true, but the authors should also indicate that in both cases is far from certain that these effects are the result of their effects on autophagy itself, as multiple publications have new questioned these relationships, and suggested that that it is from autophagy-independent functions of these molecules and/or passenger deletions that may actually be relevant. The authors should inserts some cautionary sentences that describe the results of these studies.
Response 1: I thank the reviewer for having highlighted this weakness point in the work, in fact there are many other polymorphisms and other genes involved in Crohn's disease. I decided to delete the part that could generate confusion and rewrite it as follows:
Autophagy deficiency causes inflammasome-related inflammatory diseases; genetic studies demonstrated that polymorphisms of ATG16L1 gene could be associated with Crohn’s disease, a chronic inflammatory disease of intestine [60-62]. Peripheral blood mononuclear cells from patients with Crohn’s disease expressing ATG16L1 mutant produce large quantities of IL-1b [57].
- However, some evidences suggested that inflammasome is able to regulate autophagosome formation through direct or indirect interactions between autophagic proteins (as Beclin1) and NLR domains of inflammasome sensors [58]. Unfortunately, the pathophysiology of Crohn’s disease is still under investigation; in fact it has been shown by some authors [63, 64] that other genes and predisposing environmental factors can influence the onset of the disease in patients [65, 66]. Therefore some caution must be taken in order to distinguish the real autophagy role in inflammatory bowel diseases.
Point 2: A second criticism is in the authors treatment of what is termed programmed cell death type II, which in our experience is quite rare and for the most part is referenced by papers that do not actually examine whether autophagy is actually a cause of death versus just accompanies death (since autophagy is activated in response to stressors, this is not surprising). For instance, many “papers claiming a causative role of autophagy in cell death fail to provide adequate evidence”. (See information in Cell Death Section in Guidelines for the use and interpretation of assays for monitoring autophagy, as cited by the authors.) There are legitimate studies where autophagy seems to actually be causative of cell death in physiological systems, such as in the Drosophila Salivary glands, for instance, but in our view the limited nature of genuine proof of “autophagic cell death’ is a cause for concern when review papers treat it as a well -accepted phenomenon without acknowledging the prevalence of misleading and incorrect studies. The problem deepens when further investigators continue to do flawed studies and then cite these publications as evidence for a previous foundation for their papers. It is therefore our recommendation that cautionary sentences, such as are found in Guidelines, be inserted in this section.
Response 2: The reviewer raised an important point regarding autophagic cell death. The purpose of this review is to describe the role of autophagy in the pathogenesis of diseases mainly as a survival mechanism. The concept of autophagic cell death was only mentioned because there are some works in the literature. However to improve the knowledge on causative role of autophagy in cell death, we change “programmed cell death type II” into “autophagic cell death”. We insert cautionary sentences concerning interpretation of autophagic cell death as found in Guidelines.
Some studies reported that, during Drosophila development, autophagy was involved as phisiologically relevant model of cell death, whereas apoptosis and necrosis had not activated [12]. However the term “autophagic cell death” should be used with caution and moderation. Since autophagy is a tipically pro-survival process, to prove that it has a causative role in cell death, it is necessary to have adequate evidence [13]. We must prove that other mechanism of cell death are not responsible and verify that inhibition of autophagy, by genetic or chemical means, prevents cell death. Moreover only morphological criteria are not sufficient to assess cell death; it is important to use more than one assay to measure cell death and, in particular, perform cellular viability assay [14].
Point 3: Lastly, the model for figure 4 is incorrectly illustrated as interaction between LIR containing adaptors is shown as the sole factor for recruiting these proteins to the autophagosome. While this interaction is certainly involved, adaptors, such as p62, can be recruited in the absence of LC3 (See J. Cell Biol. Vol. 192 No. 1 17–27), and thus the interaction is probably a stabilizing interaction, rather than a direct recruitment.
Response 3: Figure 4 illustrates ubiquitin-dependent or independent mechanism of selective autophagy. To avoid misunderstanding, we removed the protein receptors form the figure, and we specified in the text that p62 can be recruited in the absence of LC3.
P62, as main selective substrate, is recruited to pre-existing isolation membranes through interaction with LC3. Indeed additional studies demonstrate that p62 localizes to the autophagosome formation site on the endoplasmic reticulum without LC3 or other ATG proteins interaction (J. Cell Biol. Vol. 192 No. 1 17–27.
Reviewer 3 Report
This is a generally well-conceived and well-organized review. From this reviewer's perspective the role of autophagy in shaping immune system development, the pathogenesis and treatment of autoimmune disease, such as SLE, has been grossly overlooked beyond a single genetic association cited in reference 5.
The authors should consider discussing that autophagy is a critical pathway of immunometabolism (Trends Immunol. 39:562-576) which regulates development of the innate and adaptive immune systems, and it is profoundly altered in autoimmune inflammatory diseases in manners that can be targeted for effective treatment (Trends Immunol. 39:562-576).
Beyond a single genetic association, the process and mechanism of autophagy have been found to be skewed both in T (Ann Rheum Dis. 2014 Oct;73(10):1888-97) and B cells (Ann Rheum Dis. 2015 May;74(5):912-20) of patients with SLE. Most interestingly, autophagy is skewed in a cell-type specific manner in SLE, being enhanced in effector T cells and constrained in regulatory T cells (Arthritis Rheumatol. 70(3):427-438). The paper extensive discusses the use of hydroxychloroquine as an inhibitor of autophagy in cancer, while its role in lupus and other autoimmune diseases is ignored. Importantly, hydroxychloroquine is a mainstay of treatment in SLE and rheumatoid arthritis. Paradoxically, rapamycin was also found to be remarkably effective for treatment of SLE that involves correcting the depletion of CD8 T cells and Tregs (Lancet, 391:1186-1196) which rely on autophagy to survive (Nat Immunol. 2016 Mar;17(3):277-85). The authors should also discuss the increased prevalence of malignancy in the setting of autoimmunity and how this can be treated through regulation of autophagy.
Author Response
This is a generally well-conceived and well-organized review. From this reviewer's perspective the role of autophagy in shaping immune system development, the pathogenesis and treatment of autoimmune disease, such as SLE, has been grossly overlooked beyond a single genetic association cited in reference 5.
The authors should consider discussing that autophagy is a critical pathway of immunometabolism (Trends Immunol. 39:562-576) which regulates development of the innate and adaptive immune systems, and it is profoundly altered in autoimmune inflammatory diseases in manners that can be targeted for effective treatment (Trends Immunol. 39:562-576).
Beyond a single genetic association, the process and mechanism of autophagy have been found to be skewed both in T (Ann Rheum Dis. 2014 Oct;73(10):1888-97) and B cells (Ann Rheum Dis. 2015 May;74(5):912-20) of patients with SLE. Most interestingly, autophagy is skewed in a cell-type specific manner in SLE, being enhanced in effector T cells and constrained in regulatory T cells (Arthritis Rheumatol. 70(3):427-438).
The paper extensive discusses the use of hydroxychloroquine as an inhibitor of autophagy in cancer, while its role in lupus and other autoimmune diseases is ignored. Importantly, hydroxychloroquine is a mainstay of treatment in SLE and rheumatoid arthritis. Paradoxically, rapamycin was also found to be remarkably effective for treatment of SLE that involves correcting the depletion of CD8 T cells and Tregs (Lancet, 391:1186-1196) which rely on autophagy to survive (Nat Immunol. 2016 Mar;17(3):277-85).
The authors should also discuss the increased prevalence of malignancy in the setting of autoimmunity and how this can be treated through regulation of autophagy.
Response: I thank the reviewer for positive reply. We have carefully evaluated the suggested articles and decided to add to the work the entitled paragraph “autophagy and autoimmune diseases”.
Autoimmune diseases, as systemic lupus erythematosus (SLE), are characterized by significant changes in the prognosis and outcome of the treatments. They involve all immune system, including B and T cells of the adaptive arm, and dendritic cells, macrophages, and neutrophils of the innate arm with the consequent production of antinuclear bodies. The metabolic pathways are important regulators of the immune system differentiation and activation, so if they are altered they can lead to the development of autoimmune diseases [128]. Metabolic control of immune system differentiation starts on hematopoietic stem cells, which activate mTOR dependent autophagy [129]. mTOR is a nutrient-sensing kinase that, linking metabolic signals to genetic pattern, controls cell growth and differentation. In particular, mTOR forms two complexes: mTORC1, that drives Th1 and Th17 development, and mTORC2 that mediates Th2 development, both complexes restrict regulatory T cells (Treg) development [130]. During metabolic stress, activation of endosomal traffic by Rab4A promotes autophagy through CD4 and CD3 surface receptors, and dynamin-related protein 1 (Drp1) mitochondrial fission initiator. Lysosomal degradation of Drp1 decreases mitophagy, with consequent accumulation of large elongated mitochondria and ROS-generationg in lupus T cells [131]. ROS activate mTORC1 that cause proinflammatory necrotic death and IL-4 and IL-17 secretion by DN Tcells and depletion of CD8 memory T cells and Treg cells [132,133]. In the liver this activation precedes the production of antinuclear antibodies and disease onset in lupus-prone mice [134]. These data show a pathogenic role for Rab4-mediated Drp1 depletion and identify it as potential target for SLE treatment [131]. While autophagy has been amply examined in SLE T cells, autophagy in B cells has been less studied. Clarke and co-authors demonstrate an enhanced autophagy in murine and human lupus B cells, and that it is required for plasmablast development [135]. They demonstrated the important role of autophagy in plasmablast differentiation, because B cells isolated by ATG7 deficiency mice after in vitro stimulation failed to effectively differentiate into plasma cells. Similarly, human B cells stimulated, after autophagy inhibition, did not differentiate in plasmablasts. They found hence, before the disease onset, in a early stage of B cells of SLE mouse model the activation of autophagy that increased with age [135]. The imbalance between peripherical Th17 and Treg cells is critical to SLE pathogenesis and other autoimmune diseases [136]. Th17 cells, producing IL-17A, IL-17F and IL-22, promote the autoimmune response and favor the autoantibodies production by B cells. [137]. Treg cells control the immune homeostasis releasing IL-10 and TGF-b . The most important Treg specific transcription factor is forkhead box protein 3 (Foxp3); the expression decrease of Foxp3 in Treg cells is responsible of Treg dysfunction in SLE [138]. Considering the pivotal role of autophagy in the metabolic checkpoints and in the autoimmunity control, specifically in the SLE pathogenesis, recent studies have been addressed to autophagy targeted therapies. Some studies reported that HCQ is a mainstay of SLE and others immune diseases, as rheumatoid arthritis treatment [139]. Both Th17 and Treg cells in SLE exhibit the autophagy activated, blocking the autophagy pathway by CQ the Th17/Treg response could be rebalanced. Thus, the inhibition of autophagy reduces IL-17 secretion by reversing the Th17cells over activity observed in SLE and increases Foxp3 expression in Treg cells preserving Treg cell functionality. Thus, the immune balance is restored, lowering the serum levels of inflammatory cytokines and autoantibodies [140].Patients with SLE have Treg dysfunction due to mTOR activation. Rapamycin, acting on mTOR, inhibits Tcell proliferation and has been developed as medication under the generic designation of sirolimus [141]. In vivo study demonstrated that sirolimus abrogated autoimmuty in lupus-prone mice [142]. Recent prospective study demonstrates the efficacy and safety of sirolimus in SLE patients with severe and persistent disease [143]. Considering that SLE can be associated with an increased incidence of malignancy, mainly lymphomas, the treatment with immunosuppressive drugs could also contribute to the development of secondary cancers [144]. Emerging data suggest that rapamycin is an effective antineoplastic agent in leukemia and lymphoma, hence could be used as immunosuppressive agent for SLE patients treatment [145]. However it is not always clear if pharmacological modulation of autophagy has positive or negative effects; thus more detailed studies must be performed to develop a specific target therapy.
Round 2
Reviewer 2 Report
The revisions in the text have corrected the deficiencies in the content. English proofreading for the corrections should be done, as these corrections have introduced spelling and grammar issues in these particular sentences. Otherwise, the manuscript is now suitable for publication.
Reviewer 3 Report
The paper is timely and delivers a meaningful message.